# Association between Metabolic Syndrome and Risk of Hypopharyngeal Cancer: A Nationwide Cohort Study from Korea

**DOI:** 10.3390/cancers15184454

**Published:** 2023-09-07

**Authors:** Jeong Wook Kang, Hyeon-Kyoung Cheong, Su Il Kim, Min Kyeong Lee, Young Chan Lee, In-Hwan Oh, Young-Gyu Eun

**Affiliations:** 1Department of Otolaryngology-Head and Neck Surgery, School of Medicine, Kyung Hee University, Seoul 02447, Republic of Korea; simbody@naver.com (J.W.K.); whyracer@naver.com (S.I.K.); medchan@hanmail.net (Y.C.L.); 2Department of Internal Medicine, School of Medicine, Korea University, Ansan 02841, Republic of Korea; chongmy99@hanmail.net; 3Department of Biomedical Science and Technology, Graduate School, Kyung Hee University, Seoul 02447, Republic of Korea; msdbjar@naver.com; 4Department of Preventive Medicine, School of Medicine, Kyung Hee University, Seoul 02447, Republic of Korea

**Keywords:** hypopharyngeal cancer, metabolic syndrome, obesity, dyslipidemia, epidemiology

## Abstract

**Simple Summary:**

The incidence of hypopharyngeal cancer has been rapidly increasing, but hypopharyngeal cancer has a poorer prognosis compared to other head and neck cancers. This is because the diagnosis of hypopharyngeal cancer is late due to few symptoms in the early stages. Thus, early diagnosis and prevention are very important in the management of hypopharyngeal cancer. Metabolic syndrome, including a combination of obesity, impaired lipid and glucose metabolism, and hypertension, has recently attracted attention as a risk factor for various cancers. In this study, we investigated the association between metabolic syndrome and hypopharyngeal cancer. We confirmed that metabolic syndrome lowered the risk of hypopharyngeal cancer regardless of age, sex, smoking, and alcohol consumption. This inverse association was more prominent in women than in men.

**Abstract:**

This study evaluated the relationship between metabolic syndrome (MS) and the risk of hypopharyngeal cancer. This retrospective cohort study used data from the Korean National Health Insurance Research Database. A total of 4,567,890 participants who underwent a health checkup in 2008 were enrolled. The participants were followed until 2019, and the incidence of hypopharyngeal cancer was analyzed. We evaluated the risk of hypopharyngeal cancer according to the presence of MS, including obesity, dyslipidemia, hypertension, and diabetes, using a multivariate Cox proportional hazards model adjusted for age, sex, alcohol consumption, and smoking. During the follow-up period, 821 were newly diagnosed with hypopharyngeal cancer. MS was inversely associated with the risk of hypopharyngeal cancer (hazard ratio (HR), 0.83 [95% confidence interval (CI), 0.708–0.971]). Large waist circumference and high triglyceride levels among MS elements were both inversely related to the risk of hypopharyngeal cancer (HR: 0.82 [95% CI, 0.711–0.945] and 0.83 [95% CI, 0.703–0.978], respectively). The risk of hypopharyngeal cancer decreased with increasing comorbidity of MS in women (N = 0 vs. N = 1–2 vs. N ≥ 3; HR = 1 vs. HR = 0.511 [95% CI, 0.274–0.952] vs. HR = 0.295 [95% CI, 0.132–0.66]), but not in men. This study may improve our etiological understanding of hypopharyngeal cancer.

## 1. Introduction

Hypopharyngeal cancer has an annual incidence rate of 0.6–1.1 per 100,000 in the United States [1,2]. According to a study conducted in Korea between 1999 and 2012, the incidence of hypopharyngeal cancer is rapidly increasing among head and neck cancers [3]. Hypopharyngeal cancer has a poor prognosis and is among the worst of all head and neck subsites [4]. Hypopharyngeal cancer often presents at an advanced stage, is usually multifocal, and shows early submucosal spread [4]. The hypopharynx is rich in regional lymphatics, leading to early dissemination and cervical nodal metastasis [1,4]. Surgical treatment is difficult due to the multifocality and early lymphatic spread [4]. The prognosis of untreated hypopharyngeal cancer is poor. A recent Korean study on head and neck cancer (HNC), using data from a national patient sample cohort, found that <20% of patients survived over 12 months without treatment for hypopharyngeal cancer [5]. Owing to the difficulty of treatment and poor prognosis, prevention of hypopharyngeal cancer is important.

Effective prevention requires the elucidation of the risk factors associated with the development of hypopharyngeal cancer. Established risk factors for hypopharyngeal cancer include male sex, old age, tobacco smoking, alcohol consumption, human papillomavirus type 16, and laryngopharyngeal reflux [2,6]. Metabolic syndrome (MS) has recently attracted attention as a risk factor for other cancers, such as liver, bladder, endometrial, pancreatic, breast, rectal, and colorectal cancer [7]. MS was first described by Haller and Hanefeld in 1975 and is a cluster of risk factors for cardiovascular diseases [8]. Features of MS include a combination of obesity, impaired lipid and glucose metabolism, and hypertension. MS may have different effects on various cancers. For example, obesity acts as a risk factor for cancer, but it has been reported to have a positive effect on head and neck cancer [9,10].

However, the association between MS and the development of hypopharyngeal cancer has not yet been clearly determined. In the present study, we aimed to evaluate the association of MS with the risk of hypopharyngeal cancer. We hypothesized that an investigation of the development of hypopharyngeal cancer in a nationwide cohort database, including information about MS, would confirm the association between MS and hypopharyngeal cancer. We also analyzed that MS affects the development of hypopharyngeal cancer according to sex.

## 2. Materials and Methods

### 2.1. Data Source

This study adopted a retrospective cohort design using the Korean National Health Insurance Service (KNHIS) database. The KNHIS is a public medical insurance system operated by the Korean government [11]. The KNHIS program covers approximately 97% of the Korean population and offers free health checkups for subscribers every 1–2 years. Approximately 15 million people undergo medical checkups annually. The KNHIS shares a national health information database via an official permission process for academic research (https://nhiss.nhis.or.kr/bd/ab/bdaba000eng.do). The database includes demographic, disease, treatment, and health checkup information. Health checkup information includes data on body measurements, blood tests, tobacco smoking, and alcohol consumption. According to national policy, cancer diagnoses are quickly registered in the KHNIS database. The diagnosis of hypopharyngeal cancer was recorded using codes from the International Classification of Disease, Tenth Revision, Clinical Modification (ICD-10-CM) [12].

### 2.2. Study Population

The initial participants included 6,093,826 individuals aged >40 years who underwent health checkups provided by the KNHIS in 2008. A total of 1,525,936 individuals were excluded because of missing data, a history of HNC, a diagnosis of hypopharyngeal cancer within 12 months, age < 40 years, or a diagnosis of other head and neck cancers (Figure 1). Finally, 4,567,890 individuals were followed up from 1 year after the health checkup until 2019 to detect the development of hypopharyngeal cancer. The development of hypopharyngeal cancer was identified using the KNHIS claims records during the study period. We used C12 and C13 in ICD-10 codes.

### 2.3. Definition of Variables

Age was recorded at the time of enrollment, and body mass index (BMI) was calculated as weight (kg) divided by the square of height (m^2^). The participants were divided into four weight groups according to their BMI (underweight, <18.5; normal, 18.5–22.9; pre-obese, 23.0–24.9; obese, ≥25 kg/m^2^) [13]. Abnormal lipid profile was defined as follows: total cholesterol (TC) ≥ 200 mg/dL, high-density lipoprotein (HDL) < 40 mg/dL in men, <50 mg/dL in women, low-density lipoprotein (LDL) ≥ 130 mg/dL, or triglyceride (TG) ≥ 150 mg/dL [14]. Information on smoking and alcohol consumption was collected using a self-reported questionnaire used in the health checkups of the KNHIS. Three alcohol consumption categories were defined: none, moderate (ethanol 0–30 g/day), and heavy (ethanol ≥ 30 g/day) [15]. Smoking status was categorized as never-smoker, ex-smoker, and current smoker.

### 2.4. Metabolic Syndrome

The International Diabetes Federation criteria were used to define MS [16]. Patients with MS were defined by three of the following five criteria: (1) central obesity (waist circumference (WC) ≥ 90 cm for men or ≥80 cm for women); (2) systolic blood pressure (BP) ≥ 130 mmHg, diastolic BP ≥ 85 mmHg, or a medical history of hypertension; (3) elevated fasting blood glucose (FBG) ≥ 100 mg/dL or medical history of diabetes; (4) hypertriglyceridemia TG ≥ 150 mg/dL; and (5) low HDL cholesterol levels < 40 mg/dL for men or <50 mg/dL for women.

### 2.5. Statistical Analysis

Statistical analyses were performed using SAS software, version 9.2 (SAS Institute, Cary, NC, USA). Continuous variables were described as means and standard deviations, and categorical variables were described as frequencies and percentages. Independent *t*-tests and chi-squared tests were used for continuous and categorical variables, respectively. Three Cox regression models were used to evaluate the relative risk of hypopharyngeal cancer according to the following variables: model 1, non-adjusted model; model 2, adjusted for age and sex; and model 3, adjusted for age, sex, smoking, and alcohol consumption. The hazard ratios (HRs) and 95% confidence intervals (CIs) were calculated for all models. Statistical significance was set at *p* < 0.05.

## 3. Results

### 3.1. Baseline Characteristics

The study cohort consisted of 4,567,890 individuals (Figure 1). Of the 4,567,890 participants, 821 were newly diagnosed with hypopharyngeal cancer during the follow-up period (Table 1). The baseline characteristics of the hypopharyngeal cancer (HPC) and non-HPC groups (non-HPC) are presented in Table 1. The mean follow-up periods of HPC and non-HPC were 9.51 ± 2.66 years and 11.16 ± 1.19, respectively. The proportion of males was higher in the HPC than in the non-HPC (93% vs. 54%, *p* < 0.001). The age of the patients was higher in the HPC than in the non-HPC (61.54 ± 8.93 vs. 53.96 ± 9.32, *p* < 0.001). BMI was lower in the HPC than in the non-HPC (22.88 ± 2.86 vs. 23.98 ± 3.32, *p* < 0.001). BP and glucose levels were higher in the HPC than in the non-HPC (*p* < 0.001 and *p* < 0.001, respectively). Among the lipid profiles, TC and low-density lipoprotein (LDL) levels were lower in the HPC than in the non-HPC (*p* < 0.001 and *p* < 0.001, respectively), whereas triglyceride (TG) and HDL levels did not differ between the two groups (*p* = 0.8884 and *p* = 0.9003, respectively).

### 3.2. Effects of Established Risk Factors in Hypopharyngeal Cancer Development

Table 2 shows HRs of previously established risk factors for hypopharyngeal cancer, such as sex, age, being underweight, smoking, and alcohol consumption. Old age, male, lean body weight, smoking, and alcohol consumption were risk factors for hypopharyngeal cancer (Table 2). The HR [CI] of hypopharyngeal cancer increases significantly with age (5th decade = 1 [reference], 6th decade = 3.282 [2.547–4.23], 7th decade = 7.39 [5.757–9.487], 8th decade = 7.922 [6.031–10.406], 9th decade = 12.742 [7.526–21.573]), but decreases with BMI (model 3: underweight = 2.061 [1.529–2.778], normal = 1 [reference], pre-obese = 0.579 [0.487–0.69], obese = 0.474 [0.398–0.564]). As the frequency of unhealthy behaviors (smoking and alcohol consumption) increases, the risk of hypopharyngeal cancer also increases (model 2 HR: never smoker = 1 [reference], ex-smoker = 1.368 [1.112–1.683], current smoker = 2.268 [1.938–2.655]; no alcohol consumption = 1 [reference], mild alcohol consumption = 1.257 [1.08–1.463], heavy alcohol consumption = 2.828 [2.205–3.626]).

### 3.3. Metabolic Syndrome and Risk of Hypopharyngeal Cancer

The HRs of hypopharyngeal cancer for each component of the MS and MS-related variables are described in Table 3. The adjusted HR for hypopharyngeal cancer was 0.829 times lower in participants with MS than in those without MS (95% CI 0.708–0.971). Among the MS-related components, high WC and TG were significantly associated with a lower risk of hypopharyngeal cancer (HR [CI]: high WC = 0.82 [0.711–0.945], high TG = 0.829 [0.708–0.971]), whereas HDL, hypertension, and diabetes were not related. Similar to TG levels, TC and LDL levels were also associated with a lower risk of hypopharyngeal cancer (HR [CI]: TC = 0.819 [0.706–0.949], LDL = 0.697 [0.607–0.801]). The risk of hypopharyngeal cancer decreased in a dose-dependent manner, with an increasing number of comorbid MS conditions in women but not in men (Table 4). The risk of hypopharyngeal cancer was 0.295 times lower in females with three or more MS components than in those without.

## 4. Discussion

In this study, we found that MS was related to a decreased risk of hypopharyngeal cancer, and the preventive effect was more pronounced in women than in men. We found that known risk factors, such as male sex, old age, underweight, smoking, and alcohol consumption, were also associated with hypopharyngeal cancer.

Males have a 2–5-fold higher incidence of HNC than females, with the ratio varying by area and race [17,18]. The risk of HNC increases with age, and the median age of diagnosis was the late sixth and seventh decades of life in the United States [17]. Men and older age groups have a higher chance of developing hypopharyngeal cancer, as shown in previous studies and in our study.

Smoking and alcohol consumption are well-known risk factors for HNC and synergistically contribute to carcinogenic effects [19,20]. According to previous studies, the carcinogenic effects of tobacco are dose-dependent, and the risk of HNC is directly connected to the frequency of cigarette smoking, duration, and intensity [19,20]. In contrast, smoking cessation may reduce the risk of HNC, with research suggesting that the risk decreases with time since quitting [21,22]. Additionally, alcohol consumption independently raises the risk of HNC, particularly hypopharyngeal carcinoma [2]. This study confirmed that smoking and alcohol use enhanced the incidence of hypopharyngeal carcinoma in a dose-dependent manner.

In a large pooled epidemiologic study conducted by the International Head and Neck Cancer Epidemiology (INHANCE) Consortium, leanness was related to increased HNC risk, regardless of smoking and alcohol consumption [23]. According to a population-based study conducted in Boston, HPV-negative people are less likely to develop HNC [24]. Our finding of an inverse association between BMI and the risk of hypopharyngeal cancer was consistent with the findings of the INHANCE and Boston investigations.

This study analyzed hypertension and diabetes to identify their association with hypopharyngeal cancer risk factors (Table 3). Hypertension and diabetes were the most prevalent comorbidities among 10,524 individuals diagnosed with HNC in a previous study [25]. However, only a few studies have examined diabetes and hypertension as risk factors for hypopharyngeal cancer. A recent systematic review and meta-analysis of 14 case–control and 13 cohort studies demonstrated that type 2 diabetes was not associated with the overall risk of HNC. However, subtype analysis has revealed a minor association between type 2 diabetes and pharyngeal cancer in East Asia [26]. A recent population-based cohort study demonstrated that blood pressure had no independent effect on HNC risk [27]. Our research was undertaken in Korea (northeast Asia), and similar findings were confirmed; diabetes and hypertension were not associated with an increased risk of hypopharyngeal cancer.

However, the relationship between MS and HNC risk remains controversial. In a retrospective case–control study using the Surveillance, Epidemiology, and End Results database, Stott-Miller et al. revealed a moderately inverse association between MS and HNC [28]. MS was found to be an independent risk factor for laryngeal and oral cancer in two cohort studies [8,29]. However, a recent prospective analysis found no association between MS and the risk of HNC [27]. One case–control study also showed no association between MS and the risk of nasopharyngeal cancer [30]. Our finding that MS was inversely associated with hypopharyngeal cancer is consistent with the findings of Scott-Miller et al. Only central obesity and dyslipidemia showed an inverse association in our study, whereas Scott-Miller et al. found that all MS components were inversely associated with HNC. Numerous studies have demonstrated sex variations in MS [31]. Our findings suggest that the impact of MS may vary by sex and is more pronounced in women than in men. However, the precise reason why MS symptoms are more pronounced in women remains unclear. Given the connection between obesity and MS, it is possible that the effect of being overweight was larger in women than in males. Further research is required to clarify this point.

In general, there are a few possible mechanisms by which MS contributes to cancer development [27]. These include exposure to insulin-like growth factors, high insulin levels, and insulin resistance [32]. Chronic hyperglycemia or inflammation generates oxidative stress and consequently induces DNA damage [32]. MS is also associated with various other metabolic processes, including cytokine and chemokines [33]. Obesity is associated with both hypertrophy and hyperplasia of adipocytes. These adipocytes secrete a number of proinflammatory cytokines, such as IL-1, IL-6, and IL-8, which induce inflammation and various cancers. In addition, adiponectin secreted predominantly by white adipose tissue has beneficial antineoplastic and antiproliferative effects [34]. Adiponectin is associated with insulin sensitivity and a reduction in obesity and type 2 diabetes. Further studies of the concentration of adiponectin and other cytokines may be an important clue to finding reasons for the inverse relationship between MS and the risk of hypopharyngeal cancer. On the other hand, the mechanism by which MS prevents malignancies is not clearly known. The risk of hypopharyngeal cancer is reported to increase with poor nutrition [35]. Since malnutrition is the antithesis of MS, the inverse relationship between the risk of hypopharyngeal cancer and MS can be explained.

This study has several limitations. First, this study did not compare the characteristics of patients with and without MS. Although most cohort studies suggest baseline characteristics according to exposure status, this study did not present baseline characteristics according to MS status because a simpler study design was adopted. Second, as with most other retrospective cohort studies using an administrative database, this study could not completely rule out potential limitations such as selection biases and unmeasured confounding factors. Participants were defined as those who underwent health checkups provided by the KNHIS in 2008. Although the selection of subjects for the health checkups was almost random, selection bias cannot be completely excluded because this study did not enroll the entire Korean population. Also, lifestyles such as smoking and alcohol drinking were measured by self-reporting; the possibility of under-reporting still exists. The unmeasured confounding factor in this study was socioeconomic status. Previous studies have revealed a relationship between the risk of head and neck cancers and socioeconomic status [36]. According to previous studies, low socioeconomic status is associated with a higher risk of hypopharyngeal cancer. Considering that lower economic status is associated with a higher incidence of MS, the direction of the bias will be opposite to our results. Therefore, the effect of bias may have been limited in this study [37]. Finally, the number of women with hypopharyngeal cancer was relatively lower than that of men.

Nevertheless, to the best of our knowledge, this is the first study to comprehensively analyze the association between MS and hypopharyngeal cancer using a nationwide database. Various Cox-regression models showed that metabolic syndrome lowered the risk of hypopharyngeal cancer regardless of age, sex, smoking, and alcohol consumption. This phenomenon was more prominently observed in women than in men. Further studies of other risk factors, such as HPV infection status and laryngopharyngeal reflux, are needed to understand and prevention of hypopharyngeal cancer.

## 5. Conclusions

In conclusion, this study revealed an inverse association between MS and the risk of hypopharyngeal cancer. The inverse association was clearer in women than in men. Simultaneously, we also re-confirmed the association of established risk factors, such as sex, age, being underweight, smoking, and alcohol consumption, with an increased incidence of hypopharyngeal cancer. This study offers important insights into the risk factors for hypopharyngeal cancer.

## Figures and Tables

**Figure 1 cancers-15-04454-f001:**
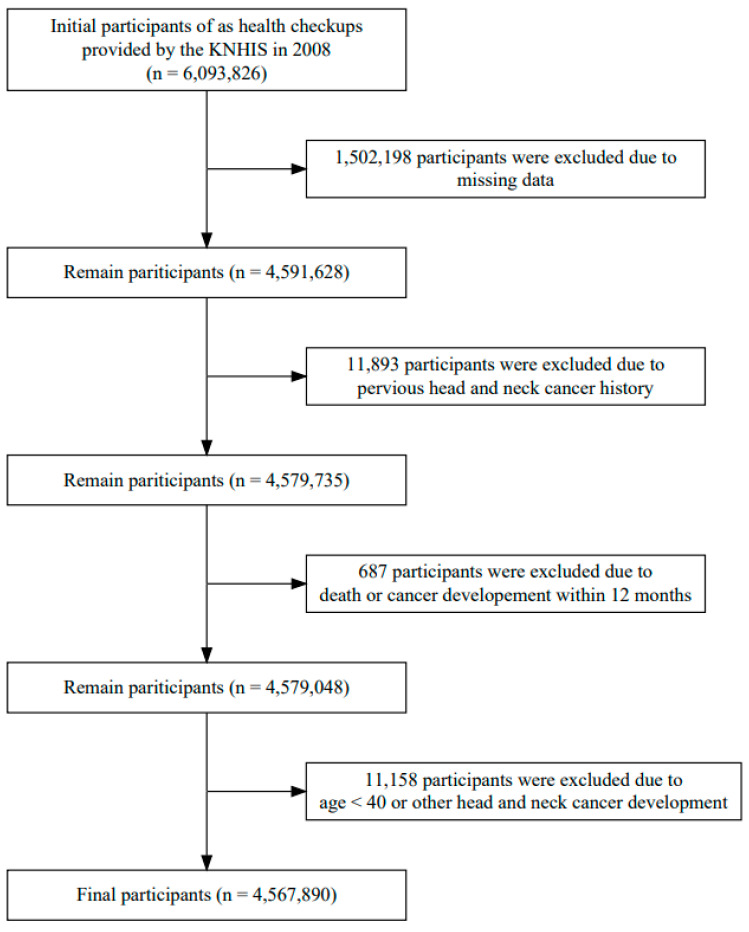
Flow chart of the studied population.

**Table 1 cancers-15-04454-t001:** Baseline characteristics of hypopharyngeal cancer and non-cancer groups.

	Hypopharyngeal Cancer	Non-Hypopharyngeal Cancer	
Variables	(n = 821)	(n = 4,567,069)	*p*-Value
Follow-up period, years	9.51 ± 2.66	11.16 ± 1.19	
Sex, n (%)			<0.001
Male	765 (93)	2,469,600 (54)	
Female	56 (7)	2,097,469 (46)	
Age, years	61.54 ± 8.93	53.96 ± 9.32	<0.001
40–49	78 (9.5)	1,753,488 (38.4)	
50–59	255 (31.1)	1,619,385 (35.5)	
60–69	307 (37.4)	780,233 (17.1)	
70–79	164 (20)	378,442 (8.3)	
80-	17 (2.1)	35,521 (0.8)	
BMI, kg/m^2^	22.88 ± 2.86	23.98 ± 3.32	<0.001
<18.5	49 (6)	87,617 (1.9)	
18.5–22.9	391 (47.6)	1,630,303 (35.7)	
23–24.9	189 (23)	1,260,472 (27.6)	
≥25	192 (23.4)	1,588,677 (34.8)	
Blood pressure, mmHg			
Systolic	128.59 ± 16.67	124.64 ± 15.63	<0.001
Diastolic	79.00 ± 10.49	77.63 ± 10.26	<0.001
Chemistry, mg/dL			
FBG	101.83 ± 25.67	98.69 ± 25.15	<0.001
TC	188.36 ± 36.56	197.45 ± 36.95	<0.001
TG	139.27 ± 89.87	139.80 ± 108.34	0.8884
HDL	55.41 ± 16.78	55.55 ± 31.89	0.9003
LDL	106.42 ± 35.68	119.64 ± 79.48	<0.001

Values are mean ± standard deviation; BMI, body mass index; FBG, fasting blood glucose; TC, total cholesterol; TG, triglyceride; HDL, high-density lipoprotein; LDL, low-density lipoprotein.

**Table 2 cancers-15-04454-t002:** Hazard ratios of potential risk factors for hypopharyngeal cancer.

		Hazard Ratio (95% Confidence Interval)		
Variables	HPC (n)	Model 1 ^a^	Model 2 ^b^	Model 3 ^c^
Sex				
Male	765	1		
Female	56	0.075 (0.057–0.098) *		
Age (years)				
40–49	78	1		
50–59	255	3.282 (2.547–4.23) *		
60–69	307	7.39 (5.757–9.487) *		
70–79	164	7.922 (6.031–10.406) *		
80-	17	12.742 (7.526–21.573) *		
BMI				
<18.5	49	2.473 (1.838–3.328) *	2.168 (1.61–2.92) *	2.061 (1.529–2.778) *
18.5–22.9	391	1	1	1
23–24.9	189	0.615 (0.517–0.732) *	0.556 (0.467–0.662) *	0.579 (0.487–0.69) *
≥25	192	0.488 (0.411–0.58) *	0.452 (0.38–0.538) *	0.474 (0.398–0.564) *
Smoking				
Never smoked	253	1	1	
Ex-smoker	90	2.81 (2.295–3.44) *	1.368 (1.112–1.683) *	
Current smoker	297	4 (3.444–4.645) *	2.268 (1.938–2.655) *	
Alcohol				
None	227	1	1	
Mild	340	2.013 (1.739–2.33) *	1.257 (1.08–1.463) *	
Heavy	73	5.121 (4.016–6.528) *	2.828 (2.205–3.626) *	

HPC, hypopharyngeal cancer; WC, waist circumference; BMI, body mass index; Asterisk (*) indicates statistical difference (*p* < 0.05). ^a^ Model 1, non-adjusted; ^b^ Model 2, adjusted for age and sex; ^c^ Model 3, adjusted for age, sex, smoking, and alcohol consumption.

**Table 3 cancers-15-04454-t003:** Hazard ratios of metabolic syndrome-related components for hypopharyngeal cancer.

		Hazard Ratio (95% Confidence Interval)		
Variables	HPC (n)	Model 1 ^a^	Model 2 ^b^	Model 3 ^c^
MS ^†^				
No	612	1	1	1
Yes	209	0.906 (0.774–1.06)	0.853 (0.729–0.999) *	0.829 (0.708–0.971) *
WC ^†^				
Low	659	1	1	1
High	162	1.125 (0.948–1.337)	0.678 (0.57–0.805) *	0.82 (0.711–0.945) *
TC				
Low	519	1	1	1
High	302	0.701 (0.608–0.808) *	0.826 (0.716–0.953) *	0.819 (0.706–0.949) *
TG ^†^				
Low	557	1	1	1
High	264	1.003 (0.866–1.161)	0.883 (0.763–1.023)	0.829 (0.703–0.978) *
HDL ^†^				
Low	693	1.612 (1.335–1.947) *	1.082 (0.894–1.308)	1.072 (0.886–1.298)
High	128	1	1	1
LDL				
Low	361	1	1	1
High	460	0.545 (0.475–0.625) *	0.675 (0.588–0.775) *	0.697 (0.607–0.801) *
HTN, Raised BP ^†^				
No	330	1	1	1
Yes	491	1.489 (1.295–1.712) *	0.98 (0.851–1.13)	0.977 (0.848–1.127)
DM, Raised FBG ^†^				
No	459	1	1	1
Yes	362	1.518 (1.322–1.742) *	1.094 (0.953–1.256)	1.08 (0.941–1.241)

HPC, hypopharyngeal cancer; MS, metabolic syndrome; WC, waist circumference; TC, total cholesterol; TG, triglyceride; HDL, high-density lipoprotein; LDL, low-density lipoprotein; BP, blood pressure; FBG, fasting blood glucose; HTN, hypertension; DM, diabetes mellitus; Asterisk (*) indicates statistical difference (*p* < 0.05); Dagger (^†^) indicates metabolic syndrome components. ^a^ Model 1, non-adjusted; ^b^ Model 2, adjusted for age and sex; ^c^ Model 3, adjusted for age, sex, smoking, and alcohol consumption.

**Table 4 cancers-15-04454-t004:** Hazard ratios of comorbid metabolic syndrome-related components according to sex.

	Hazard Ratio (95% Confidence Interval)		
	Total	Male	Female
MS components			
0	1	1	1
1–2	1.054 (0.86–1.292)	1.16 (0.932–1.443)	0.511 (0.274–0.952) *
≥3	0.866 (0.688–1.089)	0.992 (0.777–1.265)	0.295 (0.132–0.66) *
MS			
No	1	1	1
Yes	0.829 (0.708–0.971) *	0.878 (0.746–1.033)	0.488 (0.248–0.961) *
WC			
Low	1	1	1
High	0.82 (0.711–0.945) *	0.693 (0.582–0.826) *	0.558 (0.2–1.555)
TG			
Low	1	1	1
High	0.829 (0.703–0.978) *	0.83 (0.713–0.967) *	0.883 (0.479–1.627)
HDL			
Low	1.072 (0.886–1.298)	1.064 (0.868–1.303)	1.006 (0.572–1.771)
High	1	1	1
HTN			
No	1	1	1
Yes	0.977 (0.848–1.127)	1.035 (0.892–1.2)	0.522 (0.294–0.924) *
DM			
No	1	1	1
Yes	1.08 (0.941–1.241)	1.118 (0.969–1.29)	0.686 (0.367–1.282)

MS, metabolic syndrome; WC, waist circumference; TG, triglyceride; HDL, high-density lipoprotein; HTN, hypertension; DM, diabetes mellitus; asterisk (*), statistical difference (*p* < 0.05).

## Data Availability

The data that support the findings of this study are available from the KNHIS, but restrictions apply to the availability of these data, which were used under license for the current study and so are not publicly available. However, data are available from the authors upon reasonable request and with permission from KNHIS.

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
