# Peer review of "Association between Metabolic Syndrome and Risk of Hypopharyngeal Cancer: A Nationwide Cohort Study from Korea"

_cancers, 2023, doi:10.3390/cancers15184454_

Round 1

Reviewer 1 Report

Dear Authors,

Thank you so much for reviewing your article. I give you some comments to improve yours manuscript.

1.      Please describe (600 word) the Metabolic Syndrome and Risk of Hypo- 2 pharyngeal Cancer within short the novelty of your research work in an introduction section with schematic diagrams.

2.      Please add the graph of your result.

3.      Please follow your article in appropriate journals format.

4.      Make in appropriate subtitle of the article.

5.      Please check the reference of all citation.

Best Regards

Author Response

Reviewer’s comments

We would like to thank the reviewers for their helpful comments and suggestions. We have revised our manuscript in response to these comments; the following is a point-by-point response to the suggestions (bold) of the editor and reviewers (answer: blue-colored text). Revised text in the manuscript is indicated using yellow highlighting.

Reviewer 1.

Thank you so much for reviewing your article. I give you some comments to improve yours manuscript.

  1. Please describe (600 word) the Metabolic Syndrome and Risk of Hypo- 2 pharyngeal Cancer within short the novelty of your research work in an introduction section with schematic diagrams.

[Response]

Thanks for your comments. We added the ‘Simple Summary section’ before the abstract section.

Simple Summary: The incidence of hypopharyngeal cancer has been rapidly increasing, but hypopharyngeal cancer has a poorer prognosis compared to the other head and neck cancers. This is because the diagnosis of hypopharyngeal cancer is late due to few symptoms in the early stages. Thus, early diagnosis and prevention are very important in the management of hypopharyngeal cancer. Metabolic syndrome including a combination of obesity, impaired lipid and glucose metabolism, and hypertension, has recently attracted attention as a risk factor for various cancers. In this study, we investigated the association between metabolic syndrome and hypopharyngeal cancer. We confirmed that metabolic syndrome lowered the risk of hypopharyngeal cancer regardless of age, sex, smoking, and alcohol consumption. This inverse association was more prominent in women than in men.

  1. Please add the graph of your result.

[Response]

Thanks for your comments. Unfortunately, due to restrictions on access to data, we cannot add this graph now. Cox regression models, adjusted for known risk factors such as age, sex, and social history, were mainly conducted to identify independent risk factors for hypopharyngeal cancer in this study. Thus, we thought it was more prominent to use the table with hazard ratio to present the Cox analysis results in detail.

  1. Please follow your article in appropriate journals format.

[Response]

Thanks for your comments. We revised overall manuscripts according to journal format, especially in abstract section (single paragraph without heading < 200 words). Also, we added e-mail address of all authors in affiliation section.

[Previous abstract]

Background: It is unclear whether metabolic syndrome (MS) is associated with the development of hypopharyngeal cancer. Therefore, this study evaluated the relationship between MS and the risk of hypopharyngeal cancer. Methods: This retrospective cohort study used data from the Korean National Health Insurance Research Database. A total of 4,567,890 participants who underwent a health checkup in 2008 were enrolled in the study. The participants were followed until 2019, and the incidence of hypopharyngeal cancer was analyzed. We evaluated the risk of hypopharyngeal cancer according to the presence of MS, including obesity, dyslipidemia, hypertension, and diabetes, using a multivariate Cox proportional hazards model. Results: Of 4,567,890 participants, 821 were newly diagnosed with hypopharyngeal cancer during the follow-up period. The number of male patients was 765(93.2%) and of female patients was 56(6.8%). MS was inversely associated with the risk of hypopharyngeal cancer (adjusted for age, sex, alcohol consumption, smoking; hazard ratio (HR), 0.83 [95% confidence interval (CI), 0.708–0.971]). Large waist circumference and high triglyceride levels among MS elements were both inversely related to the risk of hypopharyngeal cancer (adjusted HR:0.82 [95% CI, 0.711–0.945] and 0.83 [95% CI, 0.703–0.978], respectively). Other factors associated with MS, such as hypertension, diabetes, and high-density lipoprotein cholesterol, were not associated with the development of hypopharyngeal cancer. The risk of hypopharyngeal cancer decreased with increasing comorbidity of MS in women (N=0 vs. N=1-2 vs. N ≥ 3; HR=1 vs. HR=0.511 vs. HR=0.295). Conclusion: MS is associated with a low risk of developing hypopharyngeal cancer. This relationship was more evident in women. The results of this study can be used to improve our etiological understanding of hypopharyngeal cancer.

[Revised abstract]

This study evaluated the relationship between metabolic syndrome (MS) and the risk of hypopharyngeal cancer. This retrospective cohort study used data from the Korean National Health Insurance Research Database. A total of 4,567,890 participants who underwent a health checkup in 2008 were enrolled. The participants were followed until 2019, and the incidence of hypopharyngeal cancer was analyzed. We evaluated the risk of hypopharyngeal cancer according to the presence of MS, including obesity, dyslipidemia, hypertension, and diabetes, using a multivariate Cox proportional hazards model adjusted for age, sex, alcohol consumption, smoking. During the follow-up period, 821 were newly diagnosed with hypopharyngeal cancer. MS was inversely associated with the risk of hypopharyngeal cancer (hazard ratio (HR), 0.83 [95% confidence interval (CI), 0.708–0.971]). Large waist circumference and high triglyceride levels among MS elements were both inversely related to the risk of hypopharyngeal cancer (HR: 0.82 [95% CI, 0.711–0.945] and 0.83 [95% CI, 0.703–0.978], respectively). The risk of hypopharyngeal cancer decreased with increasing comorbidity of MS in women (N=0 vs. N=1-2 vs. N ≥ 3; HR=1 vs. HR=0.511 [95% CI, 0.274-0.952] vs. HR=0.295 [95% CI, 0.132-0.66]), but not in men. This study may improve our etiological understanding of hypopharyngeal cancer.

  1. Make in appropriate subtitle of the article.

[Response]

Thanks for your comments. This journal did not require a subtitle section in the manuscript, so we added it below.

[Subtitle]

Inverse association between metabolic syndrome and hypopharyngeal cancer

  1. Please check the reference of all citation.

[Response]

Thanks for your comments. We revised re-checked all references and revised format in the manuscript.

[Examples: the location of reference numbers]

Surgical treatment is difficult due to the multifocality and early lymphatic spread. [4]

  • Surgical treatment is difficult due to the multifocality and early lymphatic spread [4].

Reviewer 2 Report

With pleasure, I read the paper titled “Association between Metabolic Syndrome and Risk of Hypopharyngeal Cancer: A Nationwide Cohort Study”. The topic is clinically relevant to clinical practice and of importance to the readers of the Cancers journal. Overall, the manuscript reads well and has good flow of ideas, up-to-date citations, and proper summary of data using tables. A major strength of the article is being among the first-ever studies to examine the relationship between MS and risk of hypopharyngeal carcinoma in depth. The rationale of the study is clearly presented in the introduction section. The research is well-articulated to encourage more research in the field. The research has some key and unavoidable limitations, all of which have been explicitly acknowledged. The conclusion is well-written. This manuscript is well-written and is very likely to be cited extensively in the future. I have the following comments:

Title. To make the title more specific and highlight its cohort origin, the authors may want to edit their title to: “Association between Metabolic Syndrome and Risk of Hypopharyngeal Cancer: A Nationwide Cohort Study from Korea”.

Abstract. Please provide the 95% CIs for the risk of hypopharyngeal cancer with respect to the increasing comorbidity of MS in women.

Introduction. Please highlight the significance of your research by indicating whether your research is the first-ever to look into the relationship between MS and risk of hypopharyngeal cancer. Also, please indicate if MS is regarded as an established risk for hypopharyngeal cancer or not. Please conclude the introduction section with a proposed hypothesis.

Methods. Was the diagnosis of hypopharyngeal cancer confirmed clinically, pathologically, subjectively by patients, or a mix of all the aforementioned? Were all of the 4,567,890 patients followed up from 2008 to 2019? For all the variables of interest (i.e., age, BMI, smoking, alcohol consumption, etc.), were these variables considered during the initial recruitment in 2008 or at the end of the follow up period 2019? For statistical analysis, have you examined whether the patient population was normally distributed or not using one of the normality tests (e.g., Kolmogorov-Smirnov or Shapiro-Wilk)? Human papillomavirus type 16 and laryngopharyngeal reflux are key risk factors for hypopharyngeal cancer; have you considered them in your Cox regression analysis?

Results. Out of curiosity, is it possible to perform the analysis of HTN based on high SBP and DBP. Also, it should be made clear if the analysis was made based on high FBG or a diagnosis of DM.

Discussion. The authors may need to provide more mechanistic/molecular reasons for the inverse relationship between MS and risk of hypopharyngeal cancer, if possible. Also, what is the relationship between MS and other ENT-related cancers, such as nasopharyngeal carcinoma, oropharyngeal carcinoma, parotid carcinoma, etc., if available. Another bias that should be acknowledged is the recall bias for history of smoking and alcohol consumption, which was reported subjectively, hence liable to overestimation or underestimation. The strength of the present study should be mentioned. For completion purposes: (a) the implications of the present study should be stated, and (2) the future directions should be described.

Author Response

Reviewer’s comments

We would like to thank the reviewers for their helpful comments and suggestions. We have revised our manuscript in response to these comments; the following is a point-by-point response to the suggestions (bold) of the editor and reviewers (answer: blue-colored text). Revised text in the manuscript is indicated using yellow highlighting.

Reviewer 2

  • With pleasure, I read the paper titled “Association between Metabolic Syndrome and Risk of Hypopharyngeal Cancer: A Nationwide Cohort Study”. The topic is clinically relevant to clinical practice and of importance to the readers of the Cancers journal. Overall, the manuscript reads well and has good flow of ideas, up-to-date citations, and proper summary of data using tables. A major strength of the article is being among the first-ever studies to examine the relationship between MS and risk of hypopharyngeal carcinoma in depth. The rationale of the study is clearly presented in the introduction section. The research is well-articulated to encourage more research in the field. The research has some key and unavoidable limitations, all of which have been explicitly acknowledged. The conclusion is well-written. This manuscript is well-written and is very likely to be cited extensively in the future. I have the following comments:

  • To make the title more specific and highlight its cohort origin, the authors may want to edit their title to: “Association between Metabolic Syndrome and Risk of Hypopharyngeal Cancer: A Nationwide Cohort Study from Korea”.

[Response]

Thanks for your comments. We revised our title following your advice.

[Previous title]

Association between Metabolic Syndrome and Risk of Hypopharyngeal Cancer: A Nationwide Cohort Study

[Revised title]

Association between Metabolic Syndrome and Risk of Hypopharyngeal Cancer: A Nationwide Cohort Study from Korea

  • Please provide the 95% CIs for the risk of hypopharyngeal cancer with respect to the increasing comorbidity of MS in women.

[Response]

Thanks for your comments. We added the corresponding 95% CIs in abstract.

[Previous abstract]

The risk of hypopharyngeal cancer decreased with increasing comorbidity of MS in women (N=0 vs. N=1-2 vs. N ≥ 3; HR=1 vs. HR=0.511 vs. HR=0.295).

[Revised abstract]

The risk of hypopharyngeal cancer decreased with increasing comorbidity of MS in women (N=0 vs. N=1-2 vs. N ≥ 3; HR=1 vs. HR=0.511 [95% CI, 0.274-0.952] vs. HR=0.295 [95% CI, 0.132-0.66]), but not in men.

  • Please highlight the significance of your research by indicating whether your research is the first-ever to look into the relationship between MS and risk of hypopharyngeal cancer. Also, please indicate if MS is regarded as an established risk for hypopharyngeal cancer or not. Please conclude the introduction section with a proposed hypothesis.

[Response]

Thanks for your comments. We revised introduction section according to your advices.

[Previous introduction]

Features of MS include a combination of obesity, impaired lipid and glucose metabolism, and hypertension. However, it is unclear whether MS is associated with the development of hypopharyngeal cancer. Therefore, this study evaluated the association of MS with the risk of hypopharyngeal cancer.

[Revised introduction (61-69 lines)]

Features of MS include a combination of obesity, impaired lipid and glucose metabolism, and hypertension. MS may have different effects on various cancers. For example, obesity acts as a risk factor for cancer, but it has been reported to have a positive effect in head and neck cancer [9,10].

However, the association between MS and development of hypopharyngeal cancer has not yet been clearly determined. In the present study, we aimed to evaluate the association of MS with the risk of hypopharyngeal cancer. We hypothesized that investigation of development of hypopharyngeal cancer in nationwide-cohort database including information about MS would confirm the association between MS and hypopharyngeal cancer. We also analyzed that MS affects the development of hypopharyngeal cancer according to sex.

  • Was the diagnosis of hypopharyngeal cancer confirmed clinically, pathologically, subjectively by patients, or a mix of all the aforementioned?

[Response]

In Korea, the diagnosis of hypopharyngeal cancer was performed pathologically through incisional or punch biopsy in all cases.

  • Were all of the 4,567,890 patients followed up from 2008 to 2019? For all the variables of interest (i.e., age, BMI, smoking, alcohol consumption, etc.), were these variables considered during the initial recruitment in 2008 or at the end of the follow up period 2019? For statistical analysis, have you examined whether the patient population was normally distributed or not using one of the normality tests (e.g., Kolmogorov-Smirnov or Shapiro-Wilk)?

[Response]

4,567,890 patients followed up from 2008. In case of the variables, age and sex is the value of 2008. Other values such as smoking, alcohol 2008 data was used. If 2008 data was missing, the data of 2009 was also used. In case of normality test, we did not measure all of the analysis because the number of subjects was large.

  • Human papillomavirus type 16 and laryngopharyngeal reflux are key risk factors for hypopharyngeal cancer; have you considered them in your Cox regression analysis?

[Response]

Thanks for your comments. We wanted to perform Cox regression models whether laryngopharyngeal reflux and Human papillomavirus (HPV) type 16 infection were an independent risk factor in hypopharyngeal cancer in a nationwide cohort study from Korea. However, this retrospective cohort data did not include HPV infection status in most cases. Moreover, the diagnosis of laryngopharyngeal reflux using only ICD-10-CM was considered inaccurate because MII-pH tests were not performed in most cases. Thus, we could not include HPV-16 infection status and laryngopharyngeal reflux in analysis of risk factors for hypopharyngeal cancer.

  • Out of curiosity, is it possible to perform the analysis of HTN based on high SBP and DBP. Also, it should be made clear if the analysis was made based on high FBG or a diagnosis of DM.

[Response]

As we described in the manuscript, HTN was defined as systolic blood pressure (BP) ≥ 130 mmHg, diastolic BP ≥ 85 mmHg, or a medical history of hypertension. DM was defined as elevated fasting blood glucose (FBG) ≥ 100 mg/dL or medical history of diabetes.

  • The authors may need to provide more mechanistic/molecular reasons for the inverse relationship between MS and risk of hypopharyngeal cancer, if possible.

[Response]

Thanks for your comments. We revised discussion section according to your advices.

[Previous introduction]

In general, there are a few possible mechanisms by which MS contributes to cancer development [27]. These include exposure to insulin-like growth factors, high insulin levels, and insulin resistance [30]. Chronic hyperglycemia or inflammation generates oxidative stress and consequently induces DNA damage [30]. On the other hand, the mechanism by which MS prevents malignancies is not clearly known. The risk of hypopharyngeal cancer is reported to increase with poor nutrition [31]. Since malnutrition is the antithesis of MS, the inverse relationship between the risk of hypopharyngeal cancer and MS can be explained.

[Revised introduction (241-256 lines)]

In general, there are a few possible mechanisms by which MS contributes to cancer development [27]. These include exposure to insulin-like growth factors, high insulin levels, and insulin resistance [32]. Chronic hyperglycemia or inflammation generates oxidative stress and consequently induces DNA damage [32]. MS is also associated with various other metabolic processes including cytokine and chemokines [33]. Obesity is associated with both hypertrophy and hyperplasia of adipocytes. These adipocytes secrete a number of proinflammatory cytokines such as IL-1, IL-6, and IL-8, which induce inflammation and various cancers. In addition, adiponectin secreted predominantly by white adipose tissue have beneficial antineoplastic and antiproliferative effects [34]. Adiponectin is associated with insulin sensitivity, and reduced in obesity and type 2 diabetes. Further studies of the concentration of adiponectin and other cytokines may be an important clue to find reasons for the inverse relationship between MS and risk of hypopharyngeal cancer. On the other hand, the mechanism by which MS prevents malignancies is not clearly known. The risk of hypopharyngeal cancer is reported to increase with poor nutrition [35]. Since malnutrition is the antithesis of MS, the inverse relationship between the risk of hypopharyngeal cancer and MS can be explained.

  • Also, what is the relationship between MS and other ENT-related cancers, such as nasopharyngeal carcinoma, oropharyngeal carcinoma, parotid carcinoma, etc., if available.

[Response]

Thanks for your comments. We added discussion section according to your advices. Laryngeal and oral cancer showed positive correlation with MS. However, nasopharyngeal cancer showed no association with MS. However, analysis of nasopharyngeal cancer was conducted through case-control study, so further analysis is required. Association between oropharyngeal cancer and MS has not yet been assessed.

[Previous discussion]

However, the relationship between MS and HNC risk remains controversial. In a ret-rospective case-control study using the Surveillance, Epidemiology, and End Results da-tabase, Stott-Miller et al. revealed a moderately inverse association between MS and HNC [28]. MS was found to be an independent risk factor for laryngeal cancer in a South Korean cohort study [8]. However, a recent prospective analysis found no association between MS and the risk of HNC [27]. Our finding that MS was inversely associated with hypopha-ryngeal cancer is consistent with the findings of Scott-Miller et al.. Only central obesity and dyslipidemia showed an inverse association in our study, whereas Scott-Miller et al. found that all MS components were inversely associated with HNC. Numerous studies have demonstrated sex variations in MS [29]. Our findings suggest that the impact of MS may vary by sex, and is more pronounced in women than in men. However, the precise reason why MS symptoms are more pronounced in women remains unclear. Given the connection between obesity and MS, it is possible that the effect of being overweight was larger in women than in males. Further research is required to clarify this point.

[Revised discussion (225-240 lines)]

However, the relationship between MS and HNC risk remains controversial. In a retrospective case-control study using the Surveillance, Epidemiology, and End Results database, Stott-Miller et al. revealed a moderately inverse association between MS and HNC [28]. MS was found to be an independent risk factor for laryngeal and oral cancer in two cohort studies [8,29]. However, a recent prospective analysis found no association between MS and the risk of HNC [27]. One case-control study also showed no association between MS and the risk of nasopharyngeal cancer [30]. Our finding that MS was inversely associated with hypopharyngeal cancer is consistent with the findings of Scott-Miller et al.. Only central obesity and dyslipidemia showed an inverse association in our study, whereas Scott-Miller et al. found that all MS components were inversely associated with HNC. Numerous studies have demonstrated sex variations in MS [31]. Our findings suggest that the impact of MS may vary by sex, and is more pronounced in women than in men. However, the precise reason why MS symptoms are more pronounced in women remains unclear. Given the connection between obesity and MS, it is possible that the effect of being overweight was larger in women than in males. Further research is required to clarify this point.

  • Another bias that should be acknowledged is the recall bias for history of smoking and alcohol consumption, which was reported subjectively, hence liable to overestimation or underestimation.

[Response]

As you mentioned, because the history of smoking and alcohol consumption is self-reported, the possibility of recall bias could not be excluded. We add it as the limitation of our study.

[Added discussion (line 266)

Also, life styles such as smoking and alcohol drinking were measured by self-reporting, possibility of under reporting still exist

  • The strength of the present study should be mentioned. For completion purposes: (a) the implications of the present study should be stated, and (2) the future directions should be described.

[Response]

Thanks for your comments. We added discussion section according to your advices.

[Added discussion (275-281 lines)]

Nevertheless, to the best of our knowledge, this is the first study to comprehensively analyze the association between MS and hypopharyngeal cancer using nationwide database. Various Cox-regression models showed that metabolic syndrome lowered the risk of hypopharyngeal cancer regardless of age, sex, smoking, and alcohol consumption. This phenomenon was more prominently observed in women than in men. Further studies of other risk factors such as HPV infection status and laryngopharyngeal reflux are needed to understanding and prevention of hypopharyngeal cancer.

Reviewer 3 Report

The study is important, findings are surprising.

Lines 72-74  The diagnosis of hypopharyngeal cancer was recorded using codes from

the International Classification of Disease, Tenth Revision, Clinical Modification (ICD-10-

CM). What ICD-10 codes were concretely used?

Table 1.  Authors provided Cox regression analyses as this is a cohort study, not case-control study. This is important to show differences between individuals with versus without MS rather than between cancer and non-cancer patients.

Authors did not write if proportional Hazard assumption was filled. This can be for example seen in Kaplan-Meier curves which must not cross.  Authors should please provide K-M curves.

Even when these additional analyses are done, readers but also authors can assume if study findings are not biased by cohort differences or statistical difficulties.

Author Response

Reviewer’s comments

We would like to thank the reviewers for their helpful comments and suggestions. We have revised our manuscript in response to these comments; the following is a point-by-point response to the suggestions (bold) of the editor and reviewers (answer: blue-colored text). Revised text in the manuscript is indicated using yellow highlighting.

Reviewer 3

The study is important, findings are surprising.

  • Lines 72-74  The diagnosis of hypopharyngeal cancer was recorded using codes from the International Classification of Disease, Tenth Revision, Clinical Modification (ICD-10-CM). What ICD-10 codes were concretely used?

[Response]

ICD-10 code used for this study is C12 and C13.

[Added Materals and Methods (line 93)]

We used C12 and C13 in ICD-10 codes.

  • Table 1.  Authors provided Cox regression analyses as this is a cohort study, not case-control study. This is important to show differences between individuals with versus without MS rather than between cancer and non-cancer patients.

[Response]

Thank you for your kind suggestion. We agree with your comments. Unfortunately, due to restrictions on access to data, additional analysis for now is hard to conduct. We hope that we could do it in the further research.

  • Authors did not write if proportional Hazard assumption was filled. This can be for example seen in Kaplan-Meier curves which must not cross.  Authors should please provide K-M curves.

[Response]

Thank you for your comments. Due to restrictions on access to data, we cannot add this graph now. Cox regression models, adjusted for known risk factors such as age, sex, and social history, were mainly conducted to identify independent risk factors for hypopharyngeal cancer in this study. Thus, we thought it was more prominent to use the table with hazard ratio to present the Cox analysis results in detail.

  • Even when these additional analyses are done, readers but also authors can assume if study findings are not biased by cohort differences or statistical difficulties.

[Response]

Thank you for reviewer’s kind comments. This study is based on the 2nd data and there are some limitations due to the characteristics of data. But this study has a large data set and long follow up period which is a great advantage.

Round 2

Reviewer 3 Report

Authors answers my comments. Unfortunately they cannot access to data used. Some problems are still present but I will click to accept